# Self-Supervised Variational Auto-Encoders

**DOI:** 10.3390/e23060747

**Published:** 2021-06-14

**Authors:** Ioannis Gatopoulos, Jakub M. Tomczak

**Affiliations:** 1Institute of Informatics, Universiteit van Amsterdam, Science Park 904, 1098 XH Amsterdam, The Netherlands; johngatop@gmail.com; 2Department of Computer Science, Vrije Universiteit Amsterdam, De Boelelaan 1111, 1081 HV Amsterdam, The Netherlands

**Keywords:** deep generative modeling, probabilistic modeling, deep learning, non-learnable transformations

## Abstract

Density estimation, compression, and data generation are crucial tasks in artificial intelligence. Variational Auto-Encoders (VAEs) constitute a single framework to achieve these goals. Here, we present a novel class of generative models, called *self-supervised Variational Auto-Encoder* (selfVAE), which utilizes deterministic and discrete transformations of data. This class of models allows both conditional and unconditional sampling while simplifying the objective function. First, we use a single self-supervised transformation as a latent variable, where the transformation is either downscaling or edge detection. Next, we consider a hierarchical architecture, i.e., multiple transformations, and we show its benefits compared to the VAE. The flexibility of selfVAE in data reconstruction finds a particularly interesting use case in data compression tasks, where we can trade-off memory for better data quality and vice-versa. We present the performance of our approach on three benchmark image data (Cifar10, Imagenette64, and CelebA).

## 1. Introduction

The framework of variational autoencoders (VAEs) provides a principled approach for learning latent-variable models. As it utilizes a meaningful low-dimensional latent space with density estimation capabilities, it forms an attractive solution for generative modeling tasks. However, its performance in terms of the test log-likelihood and quality of generated samples is often dissatisfying, and thus many modifications were proposed. In general, one can obtain a tighter lower bound and, thus, a more powerful and flexible model, by advancing over the following three components: the *encoder* [1,2,3,4], the *prior* (or *marginal* over latents) [5,6,7,8,9], and the *decoder* [10]. Recent studies have shown that, by employing deep hierarchical architectures and by carefully designing the building blocks of the neural networks, VAEs can successfully model high-dimensional data and reach state-of-the-art test likelihoods [11,12,13].

In this work, we present a novel class of VAEs, called *self-supervised Variational Auto-Encoders*, where we introduce additional variables to VAEs that result from the discrete and deterministic transformations of observed images. Since the transformations are deterministic, and they provide a specific aspect of images (e.g., contextual information through detecting edges or downscaling), we refer to them as *self-supervised representations*.

The introduction of the discrete and deterministic variables allows training deep hierarchical models efficiently by decomposing the task of learning a highly complex distribution into training smaller and conditional distributions. In this way, the model allows integrating prior knowledge about the data but still enables the synthesis of unconditional samples. Furthermore, the discrete and deterministic variables could be used to conditionally reconstruct data, which could be of great use in data compression and super-resolution tasks.

We make the following contributions: (i) We propose an extension of the VAE framework by incorporating self-supervised representations of the data. (ii) We analyze the impact of modeling natural images with different data transformations as self-supervised representations. (iii) This new type of generative model (*self-supervised Variational Auto-Encoders*), which can perform both conditional and unconditional sampling, demonstrated improved quantitative performance in terms of density estimation and the generative capabilities on image benchmarks.

## 2. Background

### 2.1. Variational Auto-Encoders

Let $x\in X^{D}$ be a vector of observable variables, where X⊆R or X⊆Z and $z\in R^{M}$ denote a vector of latent variables. We are interested in a latent variable model of the following form:(1)$$p_{\vartheta }\left(x,z\right)=p_{\theta }\left(x|z\right)\;p_{\lambda }\left(z\right).$$

For training, we require marginalizing z out; however, calculating the marginal likelihood $p_{\vartheta }\left(x\right)=\int p_{\vartheta }\left(x,z\right)dz$ is computationally intractable for non-linear stochastic dependencies. As a potential solution, a variational family of distributions could be used for approximate inference. Then, the logarithm of the marginal likelihood could be lower-bounded and the following objective function could be derived, namely, the *evidence lower bound* (ELBO) [14]:(2)$$\begin{array}{c} \ln p_{\vartheta }\left(x\right)\geq \underset{q_{\phi }\left(z|x\right)}{E}\left[\ln p_{\theta }\left(x|z\right)+\ln p_{\lambda }\left(z\right)-\ln q_{\phi }\left(z|x\right)\right] \end{array}$$

Where $q_{\phi }\left(z|x\right)$ is the variational posterior (or the *encoder*), $p_{\theta }\left(x|z\right)$ is the conditional likelihood function (or the *decoder*), $p_{\lambda }\left(z\right)$ is the *prior* (or *marginal*), and ϕ, θ, and λ denote parameters. The expectation is approximated by Monte Carlo sampling while exploiting the *reparameterization trick* to obtain unbiased gradient estimators [15]. The models are parameterized by neural networks. This generative framework is known as *Variational Auto-Encoder* (VAE) [1,15].

### 2.2. VAEs with Bijective Priors

Even though the lower-bound suggests that the prior plays a crucial role in improving the variational bounds, usually a fixed distribution is used, e.g., a standard multivariate Gaussian. While being relatively simple and computationally cheap, the fixed prior is known to result in over-regularized models that tend to ignore most of the latent dimensions [9,16,17]. Moreover, even with powerful encoders, VAEs may still fail to match the variational posterior to a unit Gaussian prior [18].

However, it is possible to obtain a rich, multi-modal prior distribution p(z) by using a *bijective* (or *flow-based*) model [19]. Formally, given a latent code z, a base distribution $p_{V}\left(v\right)$ over latent variables $v\in R^{M}$, and $f:R^{M}\overset{}{\to }R^{M}$, consisting of a sequence of *L* diffeomorphic transformations (i.e., invertible and differentiable transformations), where $f_{i}\left(v_{i-1}\right)=v_{i}$, $v_{0}=v$, and $v_{L}=z$, the *change of variable* can be used sequentially to express the distribution of z as a function of v as follows:(3)$$\log p\left(z\right)=\log p_{V}\left(v\right)-\sum_{i=1}^{L}\log\left|\frac{\partial f_{i}\left(v_{i-1}\right)}{\partial v_{i-1}}\right|,$$
where $\left|\frac{\partial f_{i}\left(v_{i-1}\right)}{\partial v_{i-1}}\right|$ is the Jacobian-determinant of the ith transformation. Thus, using the bijective prior yields the following lower-bound:(4)$$\ln p\left(x\right)\geq E_{q_{\phi }\left(z|x\right)}\left[\log p_{\theta }\left(x|z\right)-\log q_{\phi }\left(z|x\right)+\log p_{V}\left(v_{0}\right)+\sum_{i=1}^{L}\log\left|\frac{\partial f_{i}^{-1}\left(v_{i}\right)}{\partial v_{i}}\right|\right].$$

In this work, we utilize Real-Valued Non-Volume Preserving (RealNVP) transformations [19] as the prior; however, any other flow-based model could be used [3,20]. In the RealNVP, invertible transformations are composed of coupling layers. A single coupling layer, for a given input $v_{i}=\left[v_{i,a},v_{i,b}\right]$, processes only $v_{i,b}$ to obtain $v_{i+1,b}=\exp\left(s(v_{i,a})\right)⊙v_{i,b}+t\left(v_{i,a}\right)$, where s(·) and t(·) denote *scale* and *translation* neural networks, respectively. Then, the Jacobian-determinant of a single coupling layer can be calculated analytically, namely:(5)$$\left|\frac{\partial f_{i}^{-1}\left(v_{i}\right)}{\partial v_{i}}\right|=\prod_{j=1}^{D-d}\exp{\left(s\left(v_{i,a}\right)\right)}_{j}=\exp\left(\sum_{j=1}^{D-d}s{\left(x_{i,a}\right)}_{j}\right).$$

## 3. Method

### 3.1. Motivation

The idea of self-supervised learning is about utilizing original unlabeled data to create additional context information. This could be achieved in multiple manners, e.g., by adding noise to data [21] or masking data during training [22]. Self-supervised learning could also be seen as turning an unsupervised model into a supervised model by, e.g., treating predicting the next pixels as a classification task [23,24]. These are only a few examples of a quickly growing research line [25].

Here, we propose to use non-trainable transformations to obtain information about the image data. Our main hypothesis is that, since working with high-quality images is challenging, we could alleviate this problem by additionally considering partial information about them. Fitting a model to images of lower quality and then enhancing them to match the target distribution appears to be an overall easier task [26,27].

By incorporating compressed transformations (i.e., the self-supervised representations) that still contain global information, with the premise that it would be easier to approximate, the process of modeling a high-dimensional complex density can be broken down into simpler tasks. In this way, the expressivity of the model will grow and gradually result in richer, better generations.

Examples of image transformations are presented in Figure 1, such as downscaling through bicubic interpolation, blurring using Gaussian kernels, considering an n-bit image representation, edge detection, or turning an RGB image into grayscale. We notice that, with these transformations, even though they discard a great deal of information, the global structure is preserved. As a result, in practice, the model should have the ability to extract a general concept of the data and to add local information afterward. In this work, we focus on downscaling (Figure 1b–d) and edge detection or *sketching* (Figure 1i).

A positive effect of the proposed framework is that the model allows us to integrate prior knowledge through the image transformations, without losing its unconditional generative functionality. Overall, we end up with a two-level VAE with three latent variables, where one is a data transformation that can be obtained in a self-supervised fashion. In Figure 2, a schematic representation of the proposed approach with downscaling is presented.

### 3.2. Model Formulation

In our model, we consider representations that result from *deterministic* and *discrete* transformations of an image. Formally, we introduce a transformation $d:X^{D}\to X^{C}$ that takes x and returns an image representation y, e.g., a downscaled image. Since we lose information about the original image, z could be seen as a variable that compensates for lost details in x. Further, we propose to introduce an additional latent variable, $u\in R^{N}$, to model y and z.

We can define the joint distribution of x and y as p(x,y)=p(y|x)p(x), where p(y|x) becomes deterministic due to the deterministic transformation d(·). Formally, p(y|x)=δ(y−d(x)), where δ(·) denotes the Kronecker delta (for X=Z) or the Dirac delta (for X=R). Once we define the deterministic function *d*, for given data $D=\{x_{1},\ldots ,x_{N}\}$, we can obtain corresponding y’s. At first glance, it might be not apparent what is a potential advantage of our approach except for knowing how to generate pairs of data $\{{\left(x_{n},y_{n}\right\}}_{n=1}^{N}$. In order to obtain insight into that, we first consider the joint entropy of x and y, which is expressed as follows: (6)$$\begin{array}{cc} H[x,y] & =H[y|x]+H[x] \end{array}$$(7)$$\begin{array}{cc} \; & =H[x|y]+H[y]. \end{array}$$

Immediately, we can notice that, since y is a result of a deterministic function of x, the first component in Equation (Equation 6) is 0, i.e., H[y|x]=0. This follows from the fact that y is a processed version of x and contains no new information about x. As a result, we can express the entropy of x as follows:(8)H[x]=H[x|y]+H[y].

In other words, the entropy of the original data is equal to the conditional entropy of the original data and the entropy of the processed data. In general, modeling conditional distributions is an easier task than modeling the original data, and modeling p(y) may be also less complicated than modeling p(x) because y corresponds to a representation (a processed version) of x. As a result, we can turn the problem of modeling a complex marginal distribution p(x) into a self-supervised problem, where we factorize the joint distribution of x and y as p(x,y)=p(x|y)p(y). Further, we propose to compensate lost information in y by introducing z, and we also model y and z by introducing the additional latent variable u. The joint distribution over all random variables in our model is as follows:(9)p(x,y,z,u)=p(x|y,z)p(y|u)p(z|y,u)p(u).

All dependencies in our model are presented in Figure 2.

Eventually, for training, we need to calculate the logarithm of the marginal likelihood, namely, p(x,y)=∫∫p(x,y,z,u)dzu. Calculating these integrals is intractable; therefore, we again utilize the variational inference. For this purpose, we propose to apply the following family of variational distributions over latent variables:(10)Q(u,z|x,y)=q(u|y)q(z|x).

Notice that we skip a dependency between u and x. Utilizing the variational inferences gives the following lower-bound on the logarithm of the marginal likelihood:(11)$$\begin{array}{cc} \ln p\left(x,y\right)\geq E_{Q}[\ln p_{\theta }\left(x|y,z\right)+\ln p\left(z|u,y\right)+ & \ln p(y|u)+\ln p(u)+\\ \; & -\ln q(z|x)-\ln q(u|y)]. \end{array}$$

Intuitively, the premise for self-supervised Variational Auto-Encoder (selfVAE) is that the latents u will capture the global structure of the input data and the latents z will encode the missing information between y and x, guiding the model to discover the distribution of the target observations. To highlight the self-supervised part in our model, we refer to it as the *self-supervised Variational Auto-Encoder* (or selfVAE for short). Further, we propose to choose the following distributions:(12)$$\begin{array}{cc} p(v) & =N\left(v|0,1\right) \end{array}$$(13)$$\begin{array}{cc} p_{\lambda }\left(u\right) & =p\left(v\right)\;\prod_{i=1}^{F}|\det\frac{\partial f_{i}\left(v_{i-1}\right)}{\partial v_{i-1}}{|}^{-1} \end{array}$$(14)$$\begin{array}{cc} p_{\theta_{1}}\left(y|u\right) & =\sum_{i=1}^{I}\pi_{i}\left(u\right)\mathrm{Dlogistic}\left(\mu_{i}\left(u\right),s_{i}\left(u\right)\right) \end{array}$$(15)$$\begin{array}{cc} p_{\theta_{2}}\left(z|y,u\right.) & =N\left(z|\mu_{\theta_{2}}\left(y,u\right),\mathrm{diag}\left(\sigma_{\theta_{2}}\left(y,u\right)\right.)\right) \end{array}$$(16)$$\begin{array}{cc} p_{\theta_{3}}\left(x|z,y\right.) & =\sum_{i=1}^{I}\pi_{i}\left(z,y\right)\mathrm{Dlogistic}\left(\mu_{i}\left(z,y\right),s_{i}\left(z,y\right)\right) \end{array}$$(17)$$\begin{array}{cc} q_{\phi_{1}}\left(u|y\right) & =N\left(u|\mu_{\phi_{1}}\left(y\right),\mathrm{diag}\left(\sigma_{\phi_{1}}\left(y\right)\right)\right) \end{array}$$(18)$$\begin{array}{cc} q_{\phi_{2}}\left(z|x\right) & =N\left(z|\mu_{\phi_{2}}\left(x\right),\mathrm{diag}\left(\sigma_{\phi_{2}}\left(x\right)\right)\right). \end{array}$$

For the images x and y, we use Dlogistic, which is the discretized logistic distribution [28,29]. This is defined as a difference of two CDFs of the logistic distribution with a mean μ and a scale ν:(19)$$\mathrm{Dlogistic}\left(x\right)=\mathrm{sigm}\left(\frac{\left(x+0.5-\mu \right)}{\nu }\right)-\mathrm{sigm}\left(\frac{\left(x-0.5-\mu \right)}{\nu }\right),$$
where sigm is the sigmoid function, and we utilize a flow-based model for $p_{\lambda }\left(u\right.)$. We use the discretized logistic distribution because the images are represented by values between 0 and 255. For integer-valued random variables, other distributions, such as Gaussian, are inappropriate.

### 3.3. Generation and Reconstruction in selfVAE

Latent variable generative models, such as VAEs, can be used to synthesize novel content through the following process: z∼p(z)→x∼p(x|z), i.e., we first sample z, which is further fed to the conditional distribution to sample x. Additionally, these models can be utilized in reconstructing a datapoint $x^{*}$ by using the following scheme: $z∼q\left(z|x^{*}\right)\to x∼p\left(x|z\right)$. During training, VAEs use the reconstruction process.

Interestingly, our approach allows utilizing more operations regarding data generation and reconstruction. First, analogously to VAEs, the selfVAE can generate data by applying the following hierarchical sampling process (we refer to it as *generation*, see Figure 3i): $u∼p\left(u\right)\overset{}{\to }y∼p\left(y|u\right)\overset{}{\to }z∼p\left(z|u,y\right)\overset{}{\to }x∼p\left(x|y,z\right).$ In this procedure, we start with sampling u and proceed to sample z and y given u, and eventually we can obtain x. Alternatively, we can use the ground-truth y (i.e, $y^{*}=d\left(x^{*}\right)$), and sample or infer z.

Then, the generative process is the following (we refer to it as *conditional generation*, see Figure 3ii): $u∼q\left(u|y^{*}\right)\overset{}{\to }z∼p\left(z|u,y^{*}\right),\overset{}{\to }x∼p\left(x|y^{*},z\right).$ Knowing $y^{*}$ allows us to use the variational posterior to sample u instead of applying the prior p(u). This scenario is possible if we consider $y^{*}$ as a compressed version of $x^{*}$ that is shared instead of the original content.

Similarly to the conditional generation, we can conditionally reconstruct $x^{*}$ by utilizing $y^{*}$. Then, the procedure is the following (we refer to it as *conditional reconstruction*, see Figure 3iii): $z∼q\left(z|x^{*}\right)\overset{}{\to }x∼p\left(x|y^{*},z\right).$ Here, we skip inferring u, because for the given $y^{*}$ and $x^{*}$, the posterior could be used to sample z and then obtain x. If y is a result of a downscaling transformation of the input image, selfVAE is equivalent to the super-resolution as in [27].

Further, we can generate y instead of using $y^{*}$ and choose to sample or infer z. In this way, we can reconstruct an image in two ways. First, we go from $x^{*}$ to u, and then go back to x using a purely generative path (we refer to it as *reconstruction 1*, see Figure 3iv): $y^{*}=d\left(x^{*}\right)\overset{}{\to }u∼q\left(u|y^{*}\right)\overset{}{\to }y∼p\left(y|u\right)\overset{}{\to }z∼p\left(z|u,y\right)\overset{}{\to }x∼p\left(x|z,y\right).$ Alternatively, we can use the variational posterior to sample z as in the following procedure (we refer to it as *reconstruction 2*, see Figure 3v): $(y^{*}=d\left(x^{*}\right)\overset{}{\to }u∼q\left(u|y^{*}\right)\overset{}{\to }y∼p\left(y|u\right)),\;\mathrm{then}\;z∼q\left(z|x^{*}\right)\overset{}{\to }x∼p\left(x|y,z\right).$

In both reconstructions, we want to obtain u. To understand the purpose, we can again think of compression. Instead of sending high-dimensional variables, we can share the low-dimensional u and, then, infer all other variables. However, in the reconstruction 2 procedure, we must send z as well. As we will see in the experiments, this trade-off of the number of sent variables allows us to play with the compression ratio and the quality of the reconstruction.

The presented versions of generating and reconstructing images could be useful in the compression task. As we will see in the experiments, each option creates a different ratio of the reconstruction quality against the memory that we need to allocate to send information. However, every inferred variable needs to be sent; thus, more sampling corresponds to lower memory requirements. We want to highlight that, during training, we sample from variational posteriors, $z^{*}∼q\left(z|x^{*}\right),u^{*}∼p\left(u|y^{*}\right)$, and $y^{*}=d\left(x^{*}\right)$, and then we use these samples for the generative part and in the ELBO eventually. None of the procedures presented in this subsection are used for training. Here, we outline that the selfVAE allows various options for generation and reconstruction, with potential use for compression.

### 3.4. Hierarchical Self-Supervised VAE

The proposed approach can be further extended and generalized by introducing multiple transformations. For instance, we can achieve this by applying downscaling multiple times. As a result, by incorporating multiple self-supervised representations of the data, the process of modeling a high-dimensional complex density breaks down into *K* simpler modeling tasks. The resulting model is a *K*-level VAE architecture, where the overall expressivity of the model grows even further and gradually leads to generations of higher quality. It is important to realize that some transformations cannot be applied multiple times, e.g., edge detection; however, others could be used sequentially, e.g., downscaling.

Formally, we consider *K* self-supervised data transformations $d_{k}\left(\cdot \right)$ that result in *K* representations denoted by $y_{1:K}=\left[y_{1},\ldots ,y_{K}\right]$, where $y_{k}=d_{k}\left(y_{k-1}\right)$ and $y_{0}=x$. Thus, the joint distribution is factorized as follows:(20)$$p\left(x,y_{1:K}\right)=p\left(x\right)\;\prod_{k=1}^{K}\delta \left(y_{k}-d_{k}\left(y_{k-1}\right)\right).$$

However, analogously to the model with a single y, we can introduce additional latent variables for each level to compensate for lost information, $z_{k}$, and another latent variable on top, u. As a result, our model can be expressed in the following form:(21)$$\begin{array}{cc} p(x,y_{1:K},z_{1:K},u)= & p\left(x|y_{1},z_{1}\right)\;\prod_{k=1}^{K-1}\{p\left(z_{k}|y_{k},z_{k+1}\right)\times p\left(y_{k}|y_{k+1},z_{k+1}\right)\}\times \\ \; & \times p\left(z_{K}|u,y_{K}\right)\;p\left(y_{K}|u\right)\;p\left(u\right). \end{array}$$

Further, due to the intractability of the marginal likelihood function, we utilize the following family of variational distributions:(22)$$Q\left(u,z|x,y_{1:K}\right)=q\left(u|y_{K}\right)q\left(z_{1}|x\right)\prod_{k=1}^{K-1}q\left(z_{k+1}|y_{k}\right).$$

After applying the variational inference, we obtain the following objective:(23)$$\begin{array}{cc} \ln p(x,y_{1:K})\geq & E_{Q}[\ln p_{\theta }\left(x|y_{1},z_{1}\right)+\sum_{k=1}^{K-1}\{\ln p\left(z_{k}|y_{k},z_{k+1}\right)+\\ \; & +\ln p\left(y_{k}|y_{k+1},z_{k+1}\right)\}+\ln p\left(z_{K}|u,y_{K}\right)+\\ \; & +\ln p\left(y_{K}|u\right)+\ln p\left(u\right)-\ln q\left(u|y_{K}\right)+\\ \; & -\ln q\left(z_{1}|x\right)-\sum_{k=1}^{K-1}\ln q\left(z_{k+1}|y_{k}\right)]. \end{array}$$

## 4. Experiments

### 4.1. Experimental Setup

**Datasets** We evaluated the proposed model on the CIFAR-10, Imagenette64, and CelebA datasets:-*CIFAR-10* The CIFAR-10 (https://www.cs.toronto.edu/~kriz/cifar.html (accessed on 11 June 2021)) dataset is a well-known image benchmark data containing 60,000 training natural images and 10,000 validation natural images. Each image is of size 32 px × 32 px. From the training data, we put aside 15% randomly selected images as the test set. We augmented the training data by using random horizontal flips and random affine transformations and normalized the data uniformly in the range (0, 1).-*Imagenette64* Imagenette64 (https://github.com/fastai/imagenette (accessed on 11 June 2021)) is a subset of 10 easily classified classes from Imagenet https://www.image-net.org/ (accessed on 11 June 2021 ) (tench, English springer, cassette player, chain saw, church, French horn, garbage truck, gas pump, golf ball, and parachute). We downscaled the dataset to 64 px × 64 px images. Similarly to CIFAR-10, we put aside 15% randomly selected training images as the test set. We used the same data augmentation as in CIFAR-10. Please note that Imagenette64 is **not** equivalent to Imagenet64.-*CelebA* The Large-scale CelebFaces Attributes (CelebA)(http://mmlab.ie.cuhk.edu.hk/projects/CelebA.html (accessed on 11 June 2021)) dataset consists of 202,599 images of celebrities. We cropped the original images on the 40 vertical and 15 horizontal components of the top left corner of the crop box, where the height and width were cropped to 148. In addition to the uniform normalization of the image, no other augmentation was applied.**Architectures** The encoders and decoders consisted of building blocks composed of DenseNets [30], channel-wise attention [31], and ELUs [32] as activation functions. The dimensionality of all the latent variables was kept at 8×8×16=1024, and all models were trained using AdaMax [33] with data-dependent initialization [34]. Regarding the selfVAEs, in CIFAR-10, we used an architecture with a single downscaled transformation (selfVAE-downscale), while, on the remaining two datasets (CelebA and Imagenette64), we used a hierarchical three-leveled selfVAE with downscaling, and a selfVAE with sketching. Note that using sketching multiple times would result in no new data representation.All models were employed with the bijective prior (RealNVP). All models were comparable in terms of the number of parameters (the range of the weights of all models was from 32 million to 42 million). Due to our limited computational resources, these were the largest models that we were able to train. A fair comparison to the current SOTA models that use over 100 million weights [12,13] is, therefore, slightly skewed. For more details on the model architectures, please refer to the Section A.1.**Evaluation** We approximated the negative log-likelihood using 512 IW-samples [16] and express the scores in bits per dimension (*bpd*). Additionally, for CIFAR-10, we used the *Fréchet Inception Distance* (FID) [35]. We used the FID score indicating that sometimes the likelihood-based models achieved very low *bpd*; however, their perceptual quality was lower.

### 4.2. Quantitative Results

We present the results of the experiments on the benchmark datasets in Table 1. First, we notice that, on CIFAR-10, our implementation of the VAE was still lagging behind the other generative models in terms of the *bpd*; however, it was better or comparable in terms of the FID. Surprisingly, the selfVAE-downscale achieved worse *bpd* compared with the VAE with the bijective prior. A possible explanation may lie in the small image size (32×32), as the benefits of breaking down the learning process into two or more steps are not obvious given the small target dimensional space.

Nevertheless, the selfVAE-downscale achieved significantly better FID scores over the other generative models, except for the diffusion-based generative model [39]. This result could follow from the fact that downscaling allows maintaining context information about the original image, and, as a result, the general coherence is of higher quality. We want to highlight that the diffusion-based generative model proposed in [39] consists of 1000 layers; thus, comparing it to our approach is not entirely fair.

Interestingly, on the two other datasets, CelebA and Imagenette64, a three-level selfVAE-downscale achieved better *bpd* scores compared with the VAE with the bijective prior. This indicates the benefit of employing a multi-leveled self-supervised framework against the VAE in higher-dimensional data, where the plain model fails to scale efficiently. It appears that the hierarchical structure of self-supervised random variables allows encoding the missing information more efficiently in $z_{k}$, in contrast to the vanilla VAE, where all information about images must be coded in z. This result is promising and indicates that the proposed approach is of great potential for generative modeling.

Lastly, comparing the selfVAE-sketch against the hierarchical selfVAE-downscale on CelebA, we clearly see that, in terms of the *bpd*, the hierarchical selfVAE-downscale was significantly better. The selfVAE-sketch achieved an even worse *bpd* than the VAE and RealNVP. However, the generations and reconstructions (see Figure 4 and Figure A4) indicate that learning with y allowed obtaining crisp images. The only issue was learning a proper color tone (e.g., see Figure A4).

### 4.3. Qualitative Results

We present generations on CelebA in Figure 4 and on CIFAR-10 and Imagenette64 in Figure A2 (see the Appendix), and reconstructions on CIFAR-10 & CelebA in Figure 5.

We first notice that the generations from selfVAE seem to be more coherent in contrast with these from VAE, which produced, overall, more contextless and distorted generations. This result appears to be in line with the FID scores. Especially for CelebA, we observed impressive synthesis quality, great sampling diversity, and coherent generations (Figure 4). On the Imagenette64 dataset, we also observed crisper generations for our method compared to the VAE (see Figure A2 in the Appendix A.3).

Furthermore, the hierarchical selfVAE seems to be of great potential for compression purposes. In contrast to the VAE, which is restricted to using a single way of reconstructing an image, the selfVAE allows four various options with different quality/memory ratios (Figure 5). In the selfVAE-sketch, we can retrieve an image with high accuracy by using only 16% of the original data, as it manages to encode all the texture of the image to z (Figure A3). This shows the advantage of choosing prior knowledge in the learning process. Lastly, the latent variables added extra information, which defined the result, and we can alter the details of an image, such as the facial expressions (Figure 6ii, see the Appendix A.3).

In Figure 6, we visualize (i) interpolations through two ground-truth images through the latent code u and (ii) image reconstructions, where we keep the latent code u but vary all the others ($z_{1}$ and $z_{2}$) of the three-leveled selfVAE architecture. As with the previous cases, in the first case, we see that the model incorporated a rich latent space u, which is responsible for the generation and construction of the global structure of the image.

Moving from one latent code u of a given image to another, we obtained meaningful modifications of the image that resulted in images that share characteristics from both of them. However, in the latter case, we see that we can alter only the high-level features of the image when we keep the values of u but vary the others: $z_{1}$ and $z_{2}$. Interestingly, we see that, given the ground-truth image that is illustrated on the very left, we can sample different expressions and characteristics of the same person, as the latent variable u is kept constant.

## 5. Conclusions

In this paper, we showed that taking deterministic and discrete transformations resulted in coherent generations of high visual quality, and allowed integrating the prior knowledge without losing its unconditional generative functionality. The experimental results confirm that the hierarchical architectures performed better and allowed obtaining better *bpd* scores as well as better generations and reconstructions.

In the experiments, we considered two classes of image transformations, namely, *downscaling* and edge detection (*sketching*). However, there are many possible other transformations (see Figure 1), and we leave investigating them for future work. Moreover, we find the proposed approach interesting for the compression task. A similar approach with a multi-scale auto-encoder for image compression was proposed, e.g., by [40,41]. However, we still used a probabilistic framework and indicate that various non-trainable image transformations (not only multiple scales) could be of great potential.

## Figures and Tables

**Figure 1 entropy-23-00747-f001:**
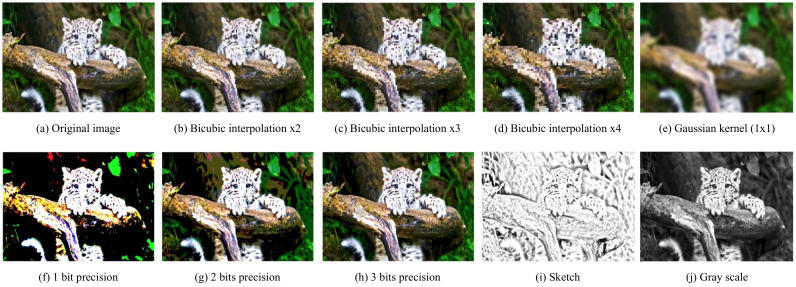
Image Transformations. All of these transformations still preserve the global structure of the samples; however, they disregard the high resolution details in different ways.

**Figure 2 entropy-23-00747-f002:**
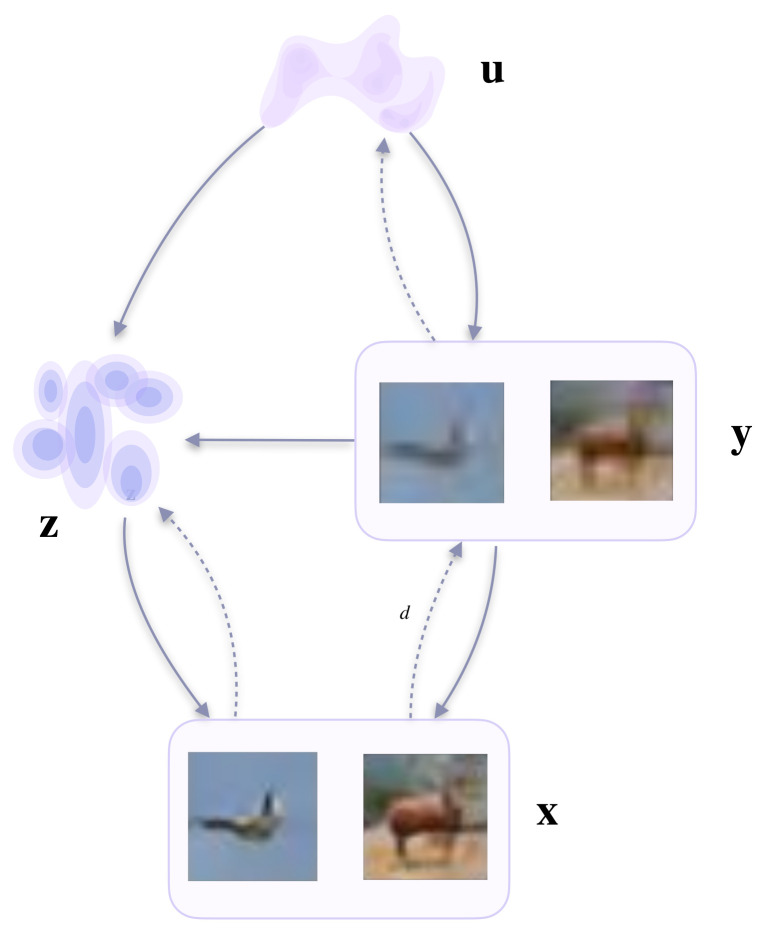
A schematic representation of the proposed approach.

**Figure 3 entropy-23-00747-f003:**
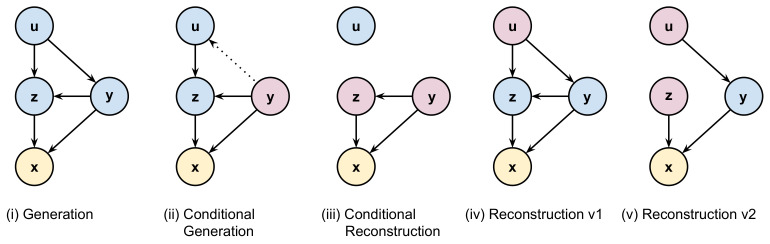
The generation and reconstruction schema in the self-supervised Variational Auto-Encoder. Blue and violet nodes represent the sampled and inferred latent codes, respectively, and yellow nodes correspond to the data.

**Figure 4 entropy-23-00747-f004:**
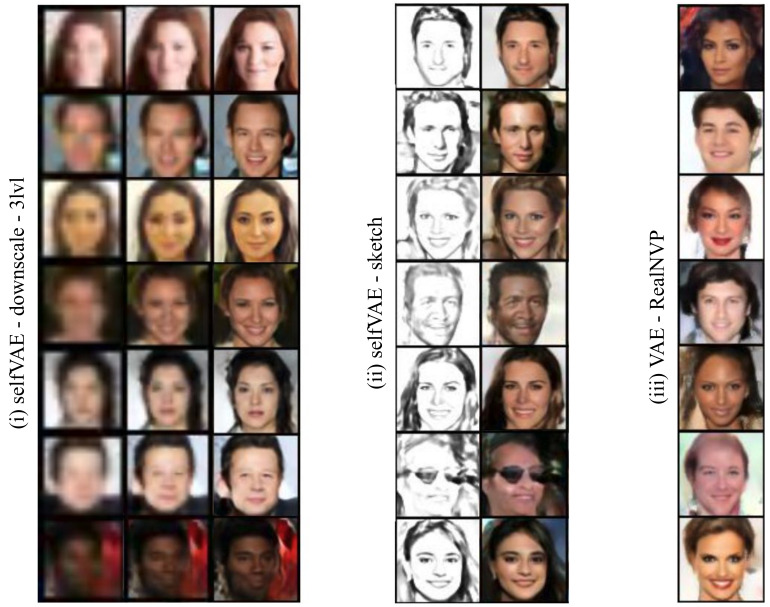
Unconditional CelebA generations from (**i**) the three-level self-supervised VAE with downscaling, (**ii**) the self-supervised VAE employed with edge detection (sketches), and (**iii**) the VAE with the RealNVP prior.

**Figure 5 entropy-23-00747-f005:**
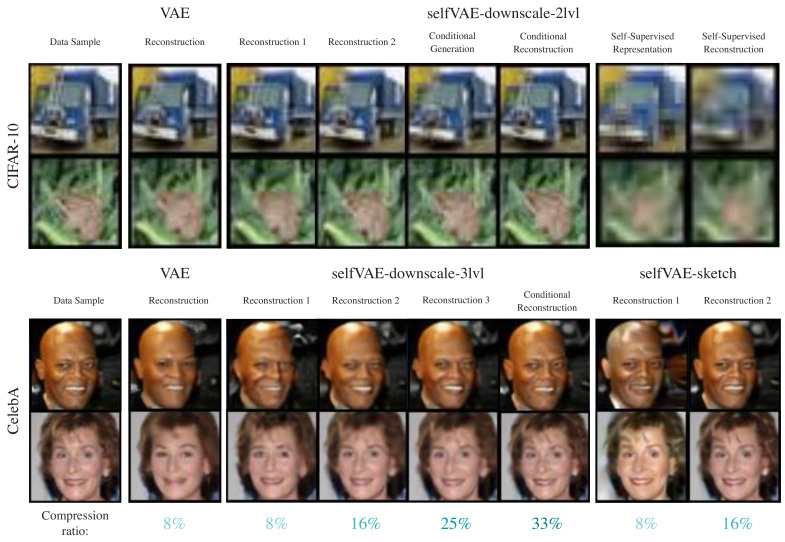
Comparison of image reconstructions with different amounts of sent information.

**Figure 6 entropy-23-00747-f006:**
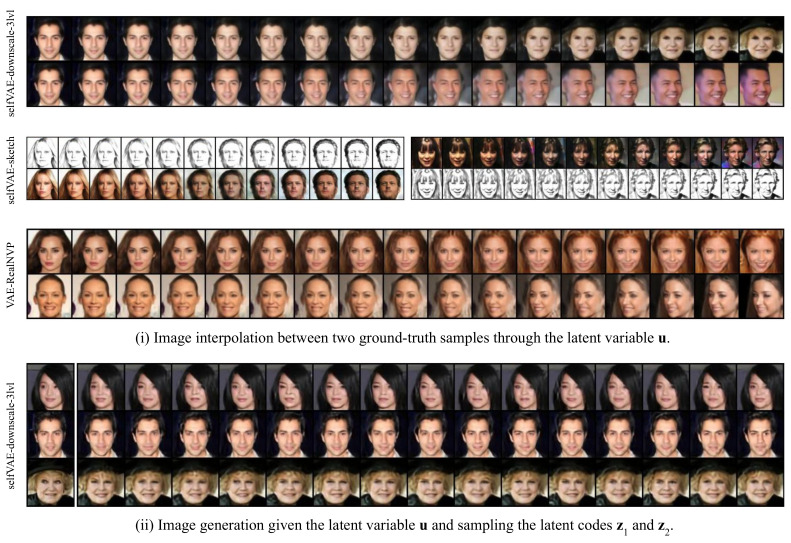
Latent space interpolations and conditional generations of the selfVAEs.

**Table 1 entropy-23-00747-t001:** Quantitative comparison on the test sets from CIFAR-10, CelebA, and Imagenette64. If available, we provide a score measured on the training set (depicted by *). Best results are in bold.

Dataset	Model	*bpd*↓	FID ↓
CIFAR-10	PixelCNN [36]	3.14	65.93
	GLOW [20]	3.35	65.93
	ResidualFlow [37]	3.28	46.37
	BIVA [12]	3.08	-
	NVAE [13]	2.91	-
	Very deep VAEs [38]	**2.87**
	DDPM [39]	3.75	**5.24** (**3.17** *)
	VAE (ours)	3.51	41.36 (37.25 *)
	selfVAE-downscale	3.65	34.71 (29.95 *)
CelebA	RealNVP [19]	3.02	-
	VAE (ours)	3.12	-
	selfVAE-sketch	3.24	-
	selfVAE-downscale-3lvl	**2.88**	-
Imagenette64	VAE (ours)	3.85	-
	selfVAE-downscale-3lvl	**3.70**	-

## Data Availability

Not applicable.

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
