# Peer review of "Self-Supervised Variational Auto-Encoders"

_entropy, 2021, doi:10.3390/e23060747_

Round 1

Reviewer 1 Report

The paper is well written and technically sound, I recommend it for publication after very minor revisions.

Please find my detailed suggestions in the following:

line 50-51: please provide an explanation or at least a reference for what concerns the parametrization trick

page 3 first line: please specify the acronym for realNVP

line 186: please provide a link and/or a reference for each dataset

line 188: "examples" of what, please specify the problem domain

line 192: please provide a reference/link and the domain of the problem

line 196: please specify which are the differences

page 8 table 1: please clarify why the measurements on the training set are available only for few experiments and/or complement the information for the remaining methods

Reviewer 2 Report

The manuscript describes one generative model that can perform conditional and unconditional sampling for data reconstruction.  
The approach seems reasonable. I have the following major comments for the article:
1. In section 4.2, the authors describe their performance as ‘significantly better than other approaches. The ‘significance’ of performance should be clarified in terms of statistical analysis. 
2. In Table 1, the FID score of the DDPM method is much lower than all rest methods. An explanation should be given to describe why this method has such superior performance according to FID metric. Particularly, the factors that affect the low FID score should be described.
3. The quantitative analysis on the comparison of the five data generation and reconstruction, as shown in Figure 3, can be added to demonstrate the effectiveness of the algorithms. 
4. In Figure 3, the description to 'circle' node and 'diamond' node should be given. 
Minors:
1. Figure 3 (vi) should be corrected to (iv)
2. Subtitle i in Figure 6 has typos. 
3. Line 183, ‘K self-super’ is duplicated.
4. Figure A1 has missing symbols. 

Round 2

Reviewer 1 Report

The authors addressed my comments, and therefore I recommend the paper for publication.

Author Response

Thank you for your comments that helped to improve our paper.

Reviewer 2 Report

The authors have addressed all my comments.

Author Response

(The authors gave the same response as above.)
